# Low Frequency Attenuation Characteristics of Two-Dimensional Hollow Scatterer Locally Resonant Phonon Crystals

**DOI:** 10.3390/ma16113982

**Published:** 2023-05-26

**Authors:** Jingcheng Xu, Changzheng Chen

**Affiliations:** School of Mechanical Engineering, Shenyang University of Technology, Shenyang 110870, China; czchen@sut.edu.cn

**Keywords:** phonon crystal, hollow scatterer, local resonance, low frequency

## Abstract

The finite element method (FEM) was applied to study the low frequency band gap characteristics of a designed phonon crystal plate formed by embedding a hollow lead cylinder coated with silicone rubber into four epoxy resin short connecting plates. The energy band structure, transmission loss and displacement field were analyzed. Compared to the band gap characteristics of three traditional phonon crystal plates, namely, the square connecting plate adhesive structure, embedded structure and fine short connecting plate adhesive structure, the phonon crystal plate of the short connecting plate structure with a wrapping layer was more likely to generate low frequency broadband. The vibration mode of the displacement vector field was observed, and the mechanism of band gap formation was explained based on the spring mass model. By discussing the effects of the width of the connecting plate, the inner and outer radii and height of the scatterer on the first complete band gap, it indicated that the narrower the width of the connecting plate, the smaller the thickness; the smaller the inner radius of the scatterer, the larger the outer radius; and the higher the height, the more conducive it is to the expansion of the band gap.

## 1. Introduction

A periodic structure consists of a number of identical structural components (called unit cells) that are joined together end-to-end and/or side-by-side to shape the whole structure. This geometry allows for creating frequency intervals in which an incident wave is not transmitted, broadly known as band gaps. Thanks to this dynamic property, a considerable amount of interest has been taken in several research fields such as structural vibrations and acoustics [1], phononics [2,3] photonics [4], and electron or plasma waves [5]. Among these, phonon crystals can effectively control the propagation of elastic waves [6]. The existence of an elastic wave band gap is the most important physical property of phonon crystals, and the band gap generating mechanism is divided into the Bragg scattering mechanism and local resonance mechanism [7]. When the elastic wavelength is on the same order of magnitude as the periodic structure size, the elastic wave is reflected back and forth between hard boundaries and cannot continue to propagate forward, resulting in a Bragg band gap. The width and location of the Bragg band gap mainly depends on the geometric parameters and shape of the scatterer and the elastic parameter difference between the scatterer and the base material [8,9]. Since the generation of the Bragg band gap is influenced by the size of the periodic structures, it is difficult to obtain in the low frequency range.

In 2000, Liu et al. [10] proposed constructing phonon crystals with locally resonant-type units for the first time. In this structure, a lead ball coated with viscoelastic material was periodically embedded into the epoxy matrix to obtain a band gap two orders of magnitude smaller than that of Bragg scattering. Since then, studies have focused on the formation mechanism and influencing factors of locally resonant band gaps [11,12,13,14,15,16]. The band gap in a locally resonant phonon crystal was generated by the coupling between the modes propagating in the matrix and the locally resonant modes of the scatterer. The group velocity of the latter was zero (flat band), and the elastic energy was completely confined within the resonator, resulting in a very narrow band gap and a sharp resonance spectrum [17]. In the study of low frequency vibration and noise control, Xiao et al. [18] analyzed the flexural vibration band gap of a thin plate with a two-dimensional three-component locally resonant structure, which is a composite structure formed by embedding a rubber-wrapped lead disk into a periodically square arranged thin epoxy resin plate. The formation of the band gap in this structure was mainly attributed to the local resonance of the rubber-wrapped resonant element. Therefore, the wrapping layer was conducive to the formation of the band gap. Li et al. [19] discussed the method of enlarging the locally resonant band gap with composite materials and proposed a composite plate type phonon crystal composed of single-sided cylindrical scatterers deposited on a two-dimensional locally resonant phonon crystal plate. The research results show that the locally resonant band gap shifted to low frequency, and the relative bandwidth significantly increased by three times compared to the single-sided locally resonant band short plate. Zhao et al. [20] studied the propagation of lamb waves in metamaterial plates dominated by bending and stretching under ultra-low density. The results show that the ultra-light metamaterial plate with textured residues deposited on its surfaces could support the strong local resonance of both symmetric and antisymmetric modes at low frequencies. The resonant frequency is very sensitive to the geometric shape of the structure. The previous study proposed a phonon crystal plate formed by periodically connecting the cylindrical scatterer coated with hard material to four epoxy resin thin plates [20]. Compared with traditional two-dimensional phonon crystal plates, this structure had a lower initial frequency of the first complete band gap. The active regulation of multiple complete band gaps was achieved after the introduction of a piezoelectric material [21].

At present, the research on the band gap characteristics of locally resonant phonon crystals mainly focuses on a phonon crystal plate composed of solid scatterers deposited on uniform thin plates, while little research has been completed on hollow scatterers with a wrapping layer [22,23,24,25,26]. To further obtain lightweight phonon crystals with more wide band gaps in the low frequency range, a new two-dimensional three-component phonon crystal plate formed by vertically embedding the hollow lead cylinder wrapped with silicone rubber into four uniform epoxy resin short plates was designed. The low frequency characteristics of the phonon crystal plate were studied using the finite element method (FEM). Its energy band structure, transmission loss and displacement field were analyzed, and the effect of the geometric parameters of the phonon crystal plate scatterer and connecting plate on the band gap was discussed in detail.

## 2. Model and Method

To obtain a wider band gap in a lower frequency range, by considering the local resonance mechanism and the influence factors of the natural frequency in the equivalent spring mass model, a new two-dimensional hollow scatterer phonon crystal plate model formed by vertically embedding a hollow lead cylinder wrapped with silicone rubber into four uniform epoxy resin short plates are designed in this paper. As shown in Figure 1, (a) is a three-dimensional view of the model, (b) is a planar view of the model and (c) is the first Brillouin zone (the shaded part is the irreducible Brillouin zone). The structural parameters of the phonon crystal plate were defined as follows: the lattice constant is a, the inner and outer radii of the hollow lead cylinder are r and r1, respectively, and the height is h1. The radius and height of the silicon rubber coating are r2 and h2, respectively, and the length and height of the epoxy resin connecting plate are l and d, respectively. The elastic material parameters required for calculations are given in Table 1, referring to [26,27].

To explore the influence of the wrapping method in the wrapping layer and the width of the connecting plate on the formation of the band gap, the contrast models are shown in Figure 2. Here, (a) is a phonon crystal plate model of a hollow lead cylinder bonded on silicone rubber embedded in the square connecting plate (expressed as contrast model 1), and (b) is a phonon crystal plate model of a hollow lead cylinder embedded in silicone rubber embedded in the square connecting plate (expressed as contrast model 2), and (c) is a phonon crystal plate model of a hollow lead cylinder bonded on a silicon rubber cylinder embedded on four short connecting plates (expressed as contrast model 3).

To analyze the propagation characteristics of elastic waves in the phonon crystal plates theoretically, the propagation equation of elastic waves is given below [3,25]:(1)$$\frac{E}{2\left(1+v\right)}\nabla^{2}u\left(r\right)+\frac{E}{2\left(1+v\right)\left(1-2v\right)}\nabla \left(\nabla \cdot u\left(r\right)\right)=-\rho \omega^{2}u\left(r\right),$$
where, *ρ* is the mass density of the medium, *E* is the Young modulus and *v* is the Poisson ratio; ***u*** is the displacement vector; ***r*** represents the position vector and ω is the angular frequency. Due to the periodicity of the structure, only the band gap characteristics of one unit cell needed to be calculated. According to Bloch’s theorem, periodic boundary conditions were applied to the four boundaries of the unit cell and the displacement field is expressed below:(2)$$u_{k}\left(r+R_{n}\right)=e^{i\left(k\cdot R_{n}\right)}u_{k}(r),$$
where, *k* represents the Bloch wave vector; and *R_n_* represents the lattice vector. By substituting Equation (2) into Equation (1), the following characteristic equation can be obtained [3]:(3)$$(K-\omega^{2}M)u=0,$$
where **K** and **M** are the global stiffness matrix and mass matrix of the lattice element, respectively. By changing the value of *k* in the first Brillouin zone and solving for the eigenvalue or frequency ω problem using the FEM, the dispersion relation and eigenmode could be obtained. To obtain the propagation modes of waves in various directions [0–360°], the wave vector k needed to vary over the entire first irreducible Brillouin zone, as shown in Figure 1c (i.e., triangular XMΓ). Γ, X and M were highly symmetric points in the Brillouin zone.

The commercial FEM software COMSOL Multiphysics 5.6 was used to calculate the dispersion relations, as it could solve the eigenvalue problems with complex boundary conditions described by Equation (2), and (3) can be solved using the eigenvalue module in COMSOL. The wave vectors were sweeping along the Γ–X–M–Γ direction in the first integrable Brillouin zone, and the eigenfrequencies and eigenvectors were calculated [3,25]. The FEM is a modern computing method that developed rapidly with the development of electronic computers. This method not only has high computational precision, but can also adapt to various complex shapes, so it is widely applied to the field of engineering computing. This paper used the finite element software to solve the energy band structure and transmission characteristics of phonon crystals.

In addition, the transmission loss spectrum was calculated to verify the numerical results of the energy band calculation. The structure used in the calculation is shown in Figure 3, where there is a finite periodic structure consisting of 20 elements in the *x*-direction. In the *y*-direction, the Bloch periodic boundary conditions were applied at two boundaries to apply the perfect matching layer to both ends of the finite structure in the *z*-direction to prevent reflection of the energy. The single frequency incident plane wave provided by the acceleration excitation source was incident from the left boundary of the finite structure, and the transmission attenuation loss *TA* is defined as below:(4)$$TA=20log\frac{d_{in}}{d_{out}},$$
where *d_in_* is the displacement acceleration excitation applied on the left side of the structure, and *d_out_* is the displacement acceleration excitation collected on the right side.

## 3. Simulation and Analysis

### 3.1. Two-Component Phononic Crystal Plate

The simplest two-dimensional phononic crystals plate is a two-component embedded structure [27,28], shown in Figure 4a. The scatterer is a lead cylinder with a radius of 8 mm, the matrix is an epoxy resin square and the length is 20 mm. The material parameters are shown in Table 1. The band structure of the two-dimensional two-component phononic crystals plate obtained by the software COMSOL 5.6 is shown in Figure 4b. The band structure shows that there are Bragg scattering bands, and the complete band gap frequency range is 41,139–44,759 Hz. The band gap of this kind of structure is dominated by Bragg scattering, which is not suitable for middle–low frequency noise reduction.

### 3.2. Calculation of Band Gap Characteristics

The values of the model structure parameters were: r = 4.5 mm, r1 = 6.5 mm, r2 = 8.0 mm, h1 = 10.0 mm, h2 = 3.0 mm, l = 5.0 mm and d = 2.0 mm. The calculation results of the energy band structure are shown in Figure 5d. According to Figure 5d, the frequency range below 1000 Hz had five complete band gaps, which are 173~582 Hz, 590~619 Hz, 620~730 Hz, 731~934 Hz and 945~958 Hz in turn. The widest band gap was the first complete band gap between the 6th and 7th bands, with a bandwidth of 420 Hz. The frequency range of the energy attenuation in the transmission loss spectrum coincided with each complete band gap of the energy band structure. The energy band structures and transmission loss spectra of the comparison models 1–3 are shown in Figure 5a–c, respectively. The lattice constants, the inner and outer radii of the scatterer, the height of the scatterer, the radius and thickness of the wrapping layer, and the thickness of the connecting plate of the three contrast models were the same as the corresponding parameters in Figure 1. The width of the connecting plate in contrast to models 1 and 2 was 20.0 mm, and the width of the connecting plate in contrast to model 3 was the same as the corresponding value in Figure 1. The calculation results showed that the three contrast models had only one complete band gap in the frequency range of less than 1000 Hz, the band gap ranges were 137~339 Hz, 260~426 Hz and 102~349 Hz, and the bandwidths were 202 Hz, 106 Hz and 247 Hz, respectively. These four structures in Figure 1 and Figure 2 showed a significant advantage over the classical structure of Figure 4 in attenuating low frequency vibrations. By comparing the energy band structure of the new two-dimensional hollow phonon crystal plate with those of the three comparison models, we found that the new two-dimensional hollow scatterer phonon crystal plate could obtain five low-frequency band gaps with a total bandwidth three times that of the conventional model.

### 3.3. Analysis of Band Gap Formation Mechanism

By comparing the energy band structures in Figure 5b,c, and combining their corresponding models, it was found that when only the connection mode of the scatterer was changed to the embedded type, that is, silicone rubber was the wrapping form, the directional band gap was widened to 7 times, indicating that the wrapping layer could greatly widen the band gap. By comparing the energy band structures in Figure 5a,d, it was found that when only the square connecting plate was changed into four small short plates, the band gap significantly increased. Therefore, reducing the width of the connecting plate was beneficial for expanding the band gap. By comparing the energy band structures in Figure 5a,d and combining their corresponding models, when the scatterer was embedded in silicone rubber, five band gaps appeared in the low frequency range, such that the width of the first complete band gap was expanded by nearly two times. Therefore, the structure with a wrapping layer could greatly increase the number and width of the band gaps.

Based on the analysis of the factors affecting the band gap in relevant literature [29,30], the eigenmodes at the upper and lower edges of the 1st to 4th complete band gaps are calculated in this paper. The corresponding eigenmodes of the marked points A–F in the energy band structure in Figure 5 are respectively shown in Figure 6A–F. The corresponding vibration mode of point A was the torsional mode of the lead cylinder, and the connecting plate remained stationary at this time. In the corresponding vibration mode of point B, the lead cylinder remained stationary and only four connecting plates vibrated along the *y*-axis, which may have been the reason for the first complete band gap termination. By comparing the corresponding vibration modes of points A and B, it can be concluded that the lower edge band of the first complete band gap was caused by the local resonance of the scatterer, while the upper edge band was induced by the vibration of the connecting plate. The vibrations of the two were completely independent without coupling, which was conducive to the regulation of the band gap. The corresponding vibration modes of point C–E and point B were basically the same, and it was the vibration of the connecting plate and had nothing to do with the scatterer. The corresponding vibration mode of point F in the high frequency region was only related to the vibration of the silicone rubber. The elastic wave was reflected back and forth between the hard boundaries, resulting in inverse overlap (i.e., Bragg scattering). Therefore, there was both the locally resonant band gap and Bragg band gap in the new two-dimensional hollow scatterer phonon crystal plate, which broke the frequency limit such that only a single band gap formation mechanism existed in the model and made it possible to realize the simultaneous regulation of high and low band gaps.

The new two-dimensional hollow scatterer phonon crystal model was equivalent to the mass spring model shown in Figure 7. The hollow lead cylinder and connecting plate were of equivalent mass, and the silicone rubber wrapping layer was the equivalent of the spring. The incident elastic wave was reflected by the surface of different media, and the phase difference between the reflected wave and the incident wave resulted in the reverse phase vibration of adjacent resonators, which were generally in equilibrium. The equivalent mass spring model frequency can be calculated by the following equation:
(5)$$f=\frac{1}{2\pi }\sqrt{\frac{K}{M}},$$
where *M* is the effective mass, *M* = *δM_i_* is the coefficient, *M_i_* is determined by the vibration part of the scatterer or connecting plate; and *K* is the stiffness coefficient of the equivalent spring.

The propagation of the elastic wave in the phonon crystal plate can be treated as the propagation of the strain energy *W*. When the vibration displacement reaches the maximum, the strain energy is fully converted into the potential energy *U* of the model. At this time, *W* = *U*, the maximum potential energy is:(6)$$U=\frac{1}{2}Kx_{max}^{2},$$
where *x*_max_ is the maximum vibration displacement, and *U*, *W* and *x*_max_ are considered dimensionless due to regularization and normalization. By calculating the strain energy *W* and the maximum vibration displacement *x*_max_ using the FEM, the equivalent stiffness *K* of the spring can be obtained based on the above equation and substituted into Equation (4) to calculate the equivalent frequency f, and then the energy band structure can be verified. In the energy band structure shown in Figure 5d, points B and D are located in the 7th and 9th frequency bands, respectively. Taking points B and D as examples, based on the corresponding vibration modes in Figure 6B,D, the frequencies at points B and D were calculated using the equation (expressed in f1) and compared with the characteristic frequencies calculated by the FEM (expressed in f2). The results are shown in Table 2. The table shows that the frequencies at points B and D calculated by the two methods were basically consistent, and the error mainly came from the neglect of the equivalent spring mass in the model, the selection of *δ* and so on.

### 3.4. Effect of Geometric Parameters on the Band Gap of Phonon Crystals

The influence of the geometric parameters in the scatterer and the connecting plate on the first complete band gap is particularly important because the generation of the first complete band gap is based on a local resonance mechanism. By analyzing the corresponding vibration modes of points A and B in the energy band structure in Figure 5d, the initial frequency of the first complete band gap was only related to the vibration of the scatterer, and the cut-off frequency was only related to the vibration of the connecting plate. Therefore, in order to specifically analyze the adjustability of the first complete band gap, the effects of the inner and outer radii and height of the scatterer and the width of the connecting plate on the first complete band gap were investigated in this paper.

In the calculation of the effect of the width in the connecting plate on the first complete band gap, the other geometric parameters were kept constant with a width of 3–20 mm and a spacing of 0.5 mm. When the width of the connecting plate was 7 mm, the initial frequency was 170 Hz, the cut-off frequency was 557 Hz and the band gap width was 387 Hz. Figure 8a shows the change in the initial and cut-off frequencies of the first complete band gap with the width in the connecting plate. According to Figure 5, this shows that the first complete band gap existed between band 6 and band 7. The analysis of the change in the distribution of the first band gap is the analysis of the change in the 6th and 7th bands. From the figure, as the width of the connecting plate increased, the 6th band (the lower boundary of the band gap) gradually moved upward and the 7th band (the upper boundary of the band gap) gradually moved downward, resulting in a gradual narrowing of the band gap width. At the width of the connecting plate of 15–17 mm, the band gap boundary changed into a concave shape. When the width of the connection plate was 16 mm, the width of the connection plate was the same as the diameter of the silicone rubber cladding and the band gap boundary was the lowest. By analyzing the equivalent mass spring model, an increase in the width of the connecting plate resulted in an increase in the vibrator mass, so the cut-off frequency of the first complete band gap decreased. As the mass of the vibrator increased, the pressure increased accordingly, which resulted in an increase in the equivalent stiffness of the silicon rubber, leading to an increase in the initial frequency of the first complete band gap. Therefore, the larger the width of the connecting plate, the narrower the band gap. Similarly, the influence of the thickness of the connecting plate on the first complete band gap was analyzed, as shown in Figure 8b. As the thickness of the connecting plate increased from 0.5 mm to 3 mm, the lower edge of the first complete band gap gradually moved upward and the upper edge moved downward, resulting in a significant narrowing of the first complete band gap. The principal analysis of the first complete band gap narrowing was the same as that of the band gap narrowing caused by the change in the width of the connecting plate.

The influence of the inner and outer radii of the scatterer on the initial and cut-off frequencies of the first complete band gap was calculated, as shown in Figure 9. In Figure 9a, as the inner radius r increased from 3 mm to 6 mm, the initial frequency increased from 125 Hz to 236 Hz. With the inner radius of the scatterer increased, the mass of the scatterer gradually decreased, which resulted in an increase in the response frequency of the vibrator, leading to a gradual increase in the initial frequency of the first complete band gap. The cut-off frequency of the first complete band gap was only related to the vibration of the connecting plate and increasing the inner radius r had no effect on the connecting plate, which is why the cut-off frequency of the first complete band gap remained unchanged. Therefore, increasing the inner radius r could reduce the width of the first complete band gap. The influence of the outer radius of the scatterer on the initial and cut-off frequencies of the first complete band gap was calculated, as shown in Figure 9b. When r1 was 5 mm, the initial frequency was 171 Hz, the cut-off frequency was 305 Hz and the bandwidth was 134 Hz. When r1 increased to 7.5 mm, the initial frequency was 244 Hz, the cut-off frequency was 1087 Hz and the bandwidth was 843 Hz. The frequency band at the lower edge of the first complete band gap gradually decreased and then increased with the increase in the outer radius, while the frequency band at the upper edge showed a trend of sharply rising. The principle is that increasing the outer radius of the scatterer can lead to an increase in the mass of the scatterer and an increase in the equivalent stiffness of the wrapping layer of silicone rubber, which can cause a rapid increase in the frequency band at the upper edge of the first complete band gap. The increase in mass was slower than the increase in equivalent stiffness, for which reason the band at the lower edge of the first complete band gap gradually decreased and then rose. Therefore, increasing the outer radius of the scatterer was beneficial to the expansion of the first complete band gap. The influence of the height of the scatterer on the initial and cut-off frequencies of the first complete band gap was analyzed, as shown in Figure 9c. The initial frequency of the first complete band gap was known to be only related to the vibration of the scatterer. Increasing the height of the scatterer would increase the mass of the vibrator, resulting in a decrease in the response frequency of the vibrator, and thus the initial frequency of the first complete band gap gradually decreased with the increase in the height of the scatterer. When the height of the scatterer increased by 10 mm, the cut-off frequency of the first complete band gap was only related to the vibration of the connecting plate, so the frequency band at the upper edge of the first complete band gap remained almost unchanged as the height of the scatterer increased.

## 4. Conclusions

In this paper, a new two-dimensional three-component phonon crystal plate model with a low bandwidth band gap formed by embedding a hollow scatterer coated with silicone rubber into four thin connecting plates was designed, and the energy band structure, transmission loss and displacement field were calculated using FEM. The calculation results showed that the structure generated several band gaps in the low frequency range (0–1000 Hz), especially a wide band gap with a width of 409 Hz in the extremely low frequency range (<600 Hz). By comparing two different models of connecting plate and wrapping layer structure, it was found that the short connecting plate structure with the wrapping layer was more conducive to the formation and expansion of band gaps. Based on the spring mass model and the displacement field, the formation mechanism of the band gap was qualitatively analyzed, including the local resonance mechanism in the low frequency range and the Bragg scattering mechanism in the high frequency range. Finally, the influence of the geometric parameters on the band gap characteristics of the model was discussed. The results showed that reducing the width of the connecting plate, decreasing the inner radius of the scatterer and increasing the outer radius of the scatterer were conducive to the expansion of the first complete band gap, and the width and position of the band gap could also be adjusted accordingly by changing the geometric parameters. In addition, the hollow scatterer structure designed in this paper can effectively reduce the mass of the whole phonon crystal plate and meet the needs of low frequency sound insulation and noise reduction in engineering while saving materials.

## Figures and Tables

**Figure 1 materials-16-03982-f001:**
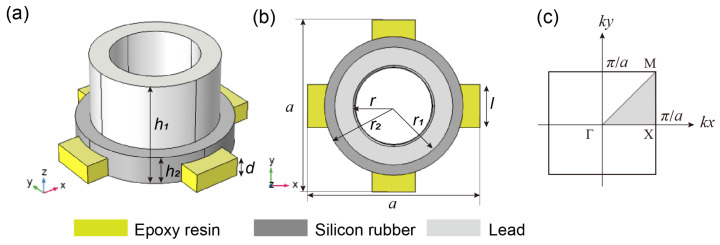
(**a**,**b**) are the two-dimensional hollow scatterer phonon crystals unit cell; (**c**) the first irreducible Brillouin zone.

**Figure 2 materials-16-03982-f002:**
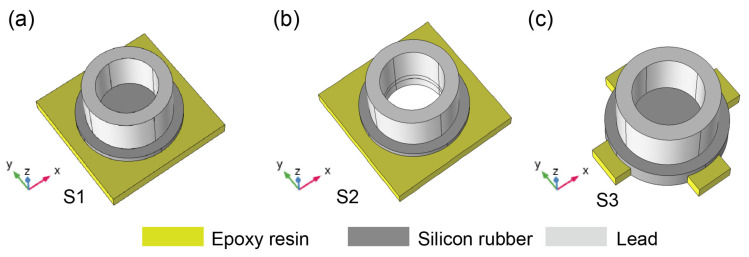
Three contrasting models of the unit cell: (**a**) a hollow lead cylinder bonded on silicone rubber embedded in the square connecting plate, (**b**) a hollow lead cylinder embedded in silicone rubber embedded in the square connecting plate, (**c**) a hollow lead cylinder bonded on a silicon rubber cylinder embedded on four short connecting plates.

**Figure 3 materials-16-03982-f003:**
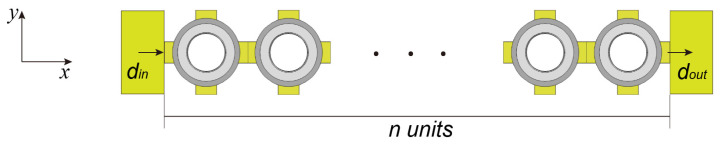
Transmission loss calculation model.

**Figure 4 materials-16-03982-f004:**
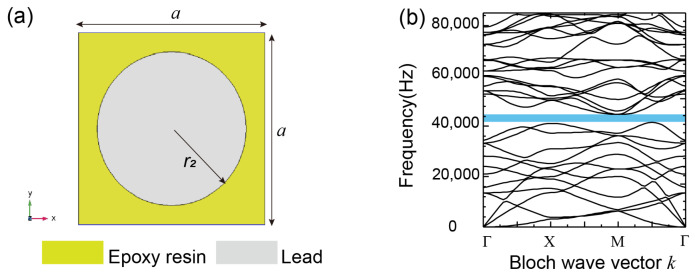
Two-dimensional two-component phononic crystals plate; (**a**) models the unit cell; (**b**) is the band structure (the black line). The blue marker is the complete band gap with a frequency range of 41,139–44,759 Hz.

**Figure 5 materials-16-03982-f005:**
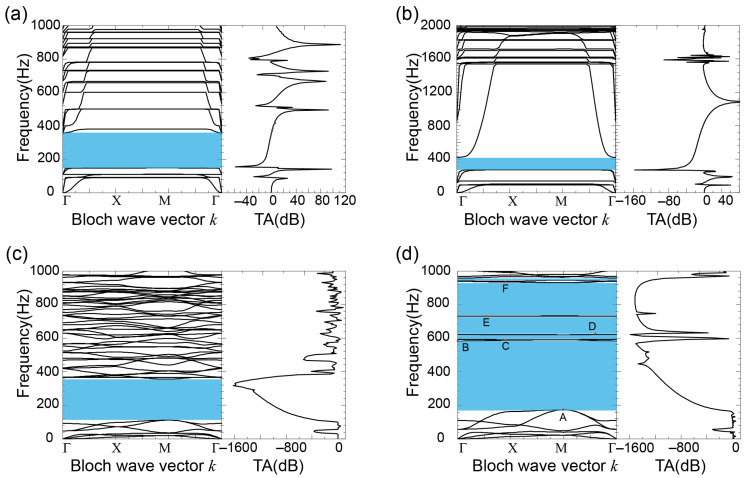
Band structure and transmission loss maps of (**a**) S1, (**b**) S2, (**c**) S3, and (**d**) hollow scatterer phonon crystals. The points A, B, C, D, E and F are complete band gap boundary points. The bands where points B, C, D, E and F are located are flat bands consistent with the characteristics of local resonance bands.

**Figure 6 materials-16-03982-f006:**
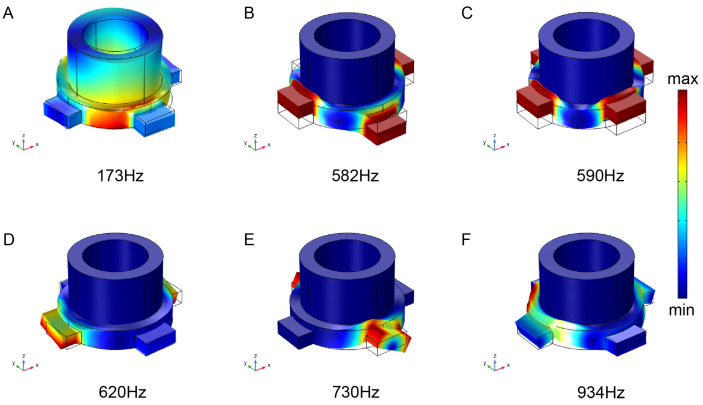
Intrinsic modes at the edge of a partial band gap of hollow scatterer phonon crystals. Corresponding to the points (**A**–**F**) in Figure 5d. The vibrations are concentrated in the connecting plate and the substrate, and the band gap formation mechanism is local resonance.

**Figure 7 materials-16-03982-f007:**
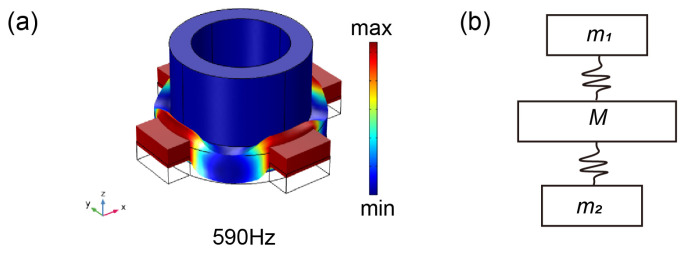
(**a**) Vibration mode; (**b**) mass spring model. *M* corresponds to the hollow scatterers and *m* corresponds to the connecting plate.

**Figure 8 materials-16-03982-f008:**
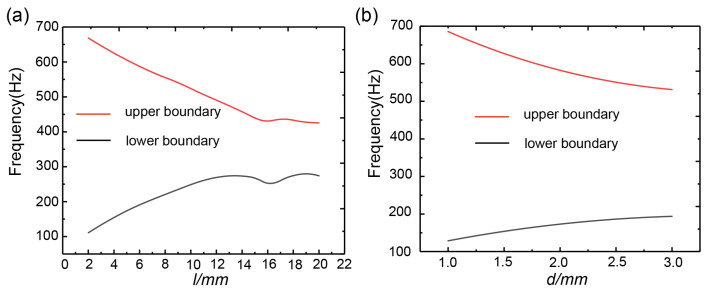
(**a**) Effect of the connecting plate width on the first complete band gap. (**b**) Effect of the connecting plate thickness on the first complete band gap.

**Figure 9 materials-16-03982-f009:**
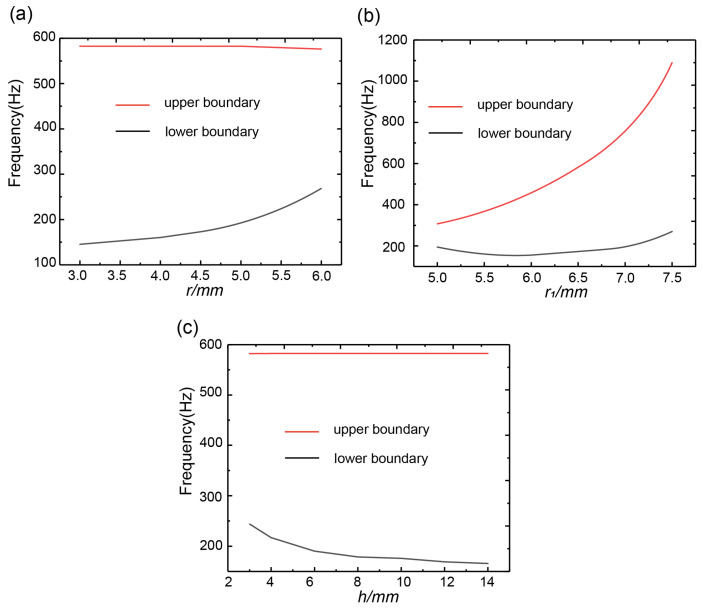
(**a**) Effect of the inner radius of the scatterer on the first complete band gap. (**b**) Effect of the outer radius of the scatterer on the first complete band gap. (**c**) Effect of the scatterer height on the first complete band gap.

**Table 1 materials-16-03982-t001:** Elastic material parameters.

Material	Density *ρ* (kg/m^3^)	Young Modulus *E* (×10^10^ Pa)	Poisson Ratio *v*
Lead	11,600	4.08	0.42
Silicon rubber	1300	1.37	0.47
Epoxy resin	1180	0.435	0.38

**Table 2 materials-16-03982-t002:** Calculation results of characteristic frequency of mass spring model.

Band	*x* _max_	*W*	*K*	*f* _1_	*f* _2_
7th	4.32	8164.96	875.02	579.44	582
9th	7.32	8697.08	324.62	590.62	620

## Data Availability

The datasets used and/or analyzed during the current study are available from the authors upon reasonable request.

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
