# Peer review of "Low Frequency Attenuation Characteristics of Two-Dimensional Hollow Scatterer Locally Resonant Phonon Crystals"

_materials, 2023, doi:10.3390/ma16113982_

Round 1
Reviewer 1 Report
In this paper, the authors propose a novel model of phonon crystal realized by embedding the hollow lead cylinder coated with silicone rubber in four short epoxy resin connection plates and studying its performance.
Several issues need to be addressed:
Main problems
1) The authors claimed to have compared the performances of the proposed structure with those of three traditional phonon crystal plates, i.e., the square connecting plate adhesive structure, the embedded structure, and the fine short connecting plate adhesive structure. However, the only comparison shown by the authors is that depicted in Fig. 4; it refers to the structures shown in Fig. 2, which are different from the three traditional structures mentioned above. Can the authors clarify this point? Furthermore, the authors should compare the performances of the proposed structure with those of more classical phonon crystal structures, such as, i.e. [1] and [2], to better elucidate its advantages.
2) The authors use the FEM method to evaluate the performances of the proposed structure. However no description of both the software used to carry out their computation and no discussion of the numerical method used to solve the generalized eigenvalue problem described by eq.(3) is provided by them.
Minor issues
1) the caption of Figure (1) “ (a) and (b) two-dimensional hollow scatterers phonon crystals unit cell; (c) the first irreducible Brillouin zone.” should be replaced with “(a) top view and (b) side view of two-dimensional hollow scatterers phonon crystals unit cell; (c) the first irreducible Brillouin zone.”
References
[1] Li, S., Chen, T., Wang, X., Li, Y., & Chen, W. (2016). Expansion of lower-frequency locally resonant band gaps using a double-sided stubbed composite phononic crystals plate with composite stubs. Physics Letters A, 380(25-26), 2167-2172.
[2] Li, L., Gang, X., Sun, Z., Zhang, X., & Zhang, F. (2018). Design of phononic crystals plate and application in vehicle sound insulation. Advances in Engineering Software, 125, 19-26.
The paper needs a minor editing of the English language.
Reviewer 2 Report
In this manuscript, “Low Frequency Attenuation Characteristics of Two-Dimensional Hollow Scatterer Locally Resonant Phonon Crystals,” FEM was applied to study the low frequency band gap characteristics of the designed phonon crystal plate formed by embedding the hollow lead cylinder coated with silicone rubber into four epoxy resin short connecting plates. Overall, this manuscript has a strong potential for another review round after applying the issues and addressing the shortcomings listed below:
1-The authors should polish/revise some grammatical mistakes and typos along the manuscript. I invite the authors to read their manuscript carefully and make the required changes where necessary.
2-Please increase size of the text provided in the figures, where necessary.
3-In the Introduction section, while discussing recent developments in the field of composite periodic structures and their applications, the following works should also be considered and cited to give a more general view to the possible readers of the work: [(i) Controlled self-assembly of plasmon-based photonic nanocrystals for high performance photonic technologies, Nano Today 37, 101072 (2021); (ii) Robust Large-Sized Photochromic Photonic Crystal Film for Smart Decoration and Anti-Counterfeiting, ACS Appl. Mater. Interfaces 14, 14618-14629 (2022)].
4-Corresponding references for the details provided in Table 1 should be provided.
5-In Figure 2, make the axis arrows a bit bigger.
6-Corresponding references for the equations should be provided.
7-Figure 5 is confusing without having a color bar. So, please put a color bar in Figure 5. Do the same for Figure 6a.
8-Please add more detailed explanation with respect to Figure 7.
N/A.
Round 2
Reviewer 1 Report
The authors have answered all my questions. Therefore, the manuscript can be accepted for publication.
The paper needs to be checked for the English language; see, for example, on page 5, lines 184-185, the statement " These 4 structures in Figures 1 and 2 show a significant advantage 183 over the classical structure Figure 4 in attenuating low frequency vibrations." should be rewritten as "These four structures, reported in Figures 1 and 2, show a significant advantage over the classical structure, reported in Figure 4, in attenuating low-frequency vibrations."
Reviewer 2 Report
Proceed for publication.